# Participation of *HHIP* Gene Variants in COPD Susceptibility, Lung Function, and Serum and Sputum Protein Levels in Women Exposed to Biomass-Burning Smoke

**DOI:** 10.3390/diagnostics10100734

**Published:** 2020-09-23

**Authors:** Alejandro Ortega-Martínez, Gloria Pérez-Rubio, Alejandra Ramírez-Venegas, María Elena Ramírez-Díaz, Filiberto Cruz-Vicente, María de Lourdes Martínez-Gómez, Espiridión Ramos-Martínez, Edgar Abarca-Rojano, Ramcés Falfán-Valencia

**Affiliations:** 1HLA Laboratory, Instituto Nacional de Enfermedades Respiratorias Ismael Cosío Villegas, Mexico City 14080, Mexico; alex_om_scv@outlook.com (A.O.-M.); glofos@yahoo.com.mx (G.P.-R.); 2Sección de Estudios de Posgrado e Investigación. Escuela Superior de Medicina, Instituto Politécnico Nacional, Plan de San Luis y Díaz Mirón s/n, Casco de Santo Tomas, Mexico City 11340, Mexico; 3Tobacco Smoking and COPD Research Department, Instituto Nacional de Enfermedades Respiratorias Ismael Cosío Villegas, Mexico City 14080, Mexico; aleravas@hotmail.com; 4Coordinación de Vigilancia Epidemiológica, Jurisdicción 06 Sierra, Tlacolula de Matamoros Oaxaca, Servicios de Salud de Oaxaca, Oaxaca 70400, Mexico; drmariel2504@hotmail.com; 5Internal Medicine Department. Hospital Civil Aurelio Valdivieso, Servicios de Salud de Oaxaca, Oaxaca 68050, Mexico; filitv6cv@hotmail.com; 6Hospital Regional de Alta Especialidad de Oaxaca, Oaxaca 71256, Mexico; ane-margo68@hotmail.com; 7Experimental Medicine Research Unit, Facultad de Medicina, Universidad Nacional Autónoma de México, Mexico City 06720, Mexico; espiri77mx@yahoo.com

**Keywords:** COPD, biomass-burning, *HHIP*, sputum supernatant, lung function, indoor pollution

## Abstract

Background: A variety of organic materials (biomass) are burned for cooking and heating purposes in poorly ventilated houses; smoke from biomass combustion is considered an environmental risk factor for chronic obstructive pulmonary disease COPD. In this study, we attempted to determine the participation of single-nucleotide variants in the *HHIP* (hedgehog-interacting protein) gene in lung function, HHIP serum levels, and HHIP sputum supernatant levels in Mexican women with and without COPD who were exposed to biomass-burning smoke. Methods: In a case-control study (COPD-BS, *n* = 186, BBES, *n* = 557) in Mexican women, three SNPs (rs13147758, rs1828591, and rs13118928) in the *HHIP* gene were analyzed by qPCR; serum and supernatant sputum protein levels were determined through ELISA. Results: The rs13118928 GG genotype is associated with decreased risk (*p* = 0.021, OR = 0.51, CI95% = 0.27–0.97) and the recessive genetic model (*p* = 0.0023); the rs1828591-rs13118928 GG haplotype is also associated with decreased risk (*p* = 0.04, OR = 0.65, CI95% 0.43–0.98). By the dominant model (rs13118928), the subjects with one or two copies of the minor allele (G) exhibited higher protein levels. Additionally, two correlations with the AG genotype were identified: BBES with FEV_1_ (*p* = 0.03, r^2^ = 0.53) and COPD-BS with FEV_1_/FVC (*p* = 0.012, r^2^ = 0.54). Conclusions: Single-nucleotide variants in the *HHIP* gene are associated with decreased COPD risk, higher HHIP serum levels, and better lung function in Mexican women exposed to biomass burning.

## 1. Introduction

Chronic obstructive pulmonary disease (COPD) is a common and tractable pathology characterized by persistent respiratory symptoms and limited airflow; these symptoms are commonly caused by significant exposure to noxious particles or gases [1]. Smoking tobacco is the principal risk factor associated with the development of COPD [2]. However, a range of organic materials (such as coal, animal dung, agricultural waste, and wood) are utilized for cooking and heating purposes in poorly ventilated houses [3], leading to chronic exposure to smoke from biomass burning. A study conducted in suburban areas near Mexico City observed that nearly 47% of women employ any biomass source for cooking and detected a COPD prevalence of 3% [4]; previously, the PLATINO study reported a COPD prevalence of 7.8% in Mexico City [5].

COPD is classified as a multifactorial disease, which means that in addition to the environmental factors strongly associated with the physiopathology of the disease, genetic factors, mostly single nucleotide polymorphisms (SNPs), have also been determined to contribute to the susceptibility and clinical variables of COPD [6]. In 2009, through genome-wide association studies (GWAS), several SNPs in the *HHIP* (hedgehog-interacting protein) gene were identified [7]; however, in this initial study, the results did not reach strict levels of significance. Finally, Wilk et al., in a GWAS, found polymorphisms in *HHIP* associated with a decrease in forced expiratory volume in the first second (FEV_1_) in the general population of the Framingham Heart Study cohort [8].

The hedgehog pathway is the signaling route in which the HHIP protein participates, and this pathway is highly conserved from an evolutionary perspective. The hedgehog signaling cascade plays an essential role in embryonic processes in vertebrates, including tooth and lung development and hair follicle anatomical structures [9]. The gene encoding the *HHIP* protein has the same name, HHIP [10], comprises 13 exons, covers approximately 91 kb, encodes a 700 amino acid protein, and is located in the 4q31.21–q31.3.9 gene region [11].

In this study, we attempted to determine the participation of single-nucleotide variants in the *HHIP* gene in lung function, HHIP serum levels, and HHIP sputum supernatant levels in Mexican women with and without COPD who were exposed to biomass-burning smoke.

## 2. Materials and Methods

### 2.1. Case and Control Groups

Seven hundred and forty-three Mexican women were included in a case-control study. These subjects attended the COPD clinic, which is part of the Department of Smoking and COPD Research of the Instituto Nacional de Enfermedades Respiratorias Ismael Cosio Villegas (INER), Mexico.

Applying diagnostic criteria according to the Global Initiative for Chronic Obstructive Lung Disease (GOLD) recommendations [12] and considering the symptoms and the deterioration of the patient’s health status, a team of pulmonary specialists completed the clinical evaluation. The diagnosis was confirmed using lung function tests (by post-bronchodilator spirometry), considering a ratio of forced expiratory volume in the first second/forced vital capacity (FEV_1_/FVC) < 70% to be proof of COPD according to the reference values for Mexicans reported by Perez-Padilla et al. [13].

Women who employed firewood as an organic fuel source (biomass) for indoor cooking, were older than 40 years, were directly exposed to biomass-burning smoke, had an accumulated biomass-burning smoke exposure index (BBEI, calculated as the average number of hours spent cooking daily per the total number of years exposed) higher than 100 h/year for biomass smoke, exhibited FEV_1_/FVC < 70%, were never smokers and were never exposed to second-hand tobacco smoke or other fumes or gases associated with COPD development were classified into the COPD-BS (*n* = 186) group.

All included patients were clinically stable, were not utilizing supplementary oxygen at the enrollment time, did not have a history of previous exacerbations, and had not been administered antibiotics or systemic corticosteroid treatments for at least three months. Consecutive COPD patients were enrolled from the COPD support clinic from 2015 to 2019. Additionally, GOLD stages I and II were grouped as G1, while stages III and IV were grouped as G2.

The control group consisted of participants who had been exposed to biomass-burning smoke (BBES, *n* = 557) and did not have COPD, including those with normal spirometry parameters (FEV_1_/FVC ≥ 70%) and without a history of active or passive smoking or non-COPD respiratory or chronic inflammatory diseases.

All participants were part of the national program to achieve equality between women and men through the Early Diagnosis/Breath Without Smoke campaigns for women living in rural areas, primarily in the northern highlands of the state of Oaxaca and suburban areas of the Tlalpan mayoralty of Mexico City.

All participants fulfilled a family questionnaire regarding inherited pathologies, by which participants who reported suffering some pulmonary or chronic inflammatory disease were excluded, as well as those with ancestry different from Mexican (that is, with no Mexican-by-birth parents and grandparents). Participants had no biological relations among themselves or with the subjects in the corresponding comparison group, and they had no history of family pulmonary diseases.

### 2.2. Ethics Approval and Informed Consent

This study was reviewed and accepted by the Institutional Committees for Investigation, Ethics in Research, and Biosecurity of the Instituto Nacional de Enfermedades Respiratorias Ismael Cosío Villegas (INER) (approbation number: B11–19). All participants were informed of the protocol′s aims after being given a detailed description of the study and being invited to participate as volunteers. All of the participants signed an informed consent paper and were supplied with a privacy statement describing the legal protection of personal data; both documents were approved (14 May 2019) by the Institutional Research and Ethics in Research Committees. All analyses were conducted following the relevant guidelines and regulations. The STREGA (STrengthening the REporting of Genetic Association) [14] recommendations were taken into consideration in the design of this genetic association study.

### 2.3. Blood Sample Processing and DNA Extraction

The sample processing began with a whole-blood 15 mL blood sample obtained by venipuncture and collected in two EDTA tubes (S-Monovette 4.9 mL K3E, Sarstedt, Nümbrecht, Germany) and another tube for obtaining serum (S-Monovette 4.9 mL Z-Gel, Sarstedt, Nümbrecht, Germany), and subsequent centrifugation was employed for 5 min at 4500 rpm to separate the peripheral blood mononuclear cells (PBMCs) and serum. Samples were stored in cryopreservation tubes at −80 °C until use.

### 2.4. Sputum Induction and Sample Preparation

Based on genotype analysis, we selected a subsample of participants for more in-depth characterization. To obtain sputum, we followed a previously published protocol [15]; briefly, participants were treated with a nebulizer with a sterile 7% saline solution. Treatment lasted for 5 min followed by a rest period of 5 min. Treatment and rest cycles were repeated three times.

The sample was mechanically disaggregated using 1X PBS buffer (Invitrogen; Carlsbad, CA, USA) in equal volumes to eliminate excess mucus followed by centrifugation at 4500 rpm for 10 min, and the saliva was extracted. Then, 10 mL of sterile 0.9% saline solution was added, and the sample was centrifuged again at 4500 rpm for 10 min, and the supernatant was separated into 1.8 mL aliquots. These aliquots were concentrated using a SpeedVac Concentrator (Thermo Fisher Scientific, Asheville NC, USA) at 14,000 rpm for 12 h, resuspended in 1 mL of 1X PBS and stored at −80 °C until use.

### 2.5. SNP Selection

SNPs were selected based on a bibliographic search in the National Center for Biotechnology Information (NCBI) database, identifying polymorphisms previously associated with COPD in different GWAS analyses and having been positively replicated in at least two other populations. Additionally, we considered a minor allele frequency (MAF) higher than 5% in the Mexican population in Los Angeles according to the 1000 Genomes Project [16]. Appendix A shows all selected SNPs.

### 2.6. SNP Genotyping

The allele discrimination of SNP variants was performed using commercial TaqMan probes (Applied Biosystems, CA, USA) at a 20X concentration. We selected three SNPs: rs13118928 (commercial probe id: C__11375931_20), rs1828591 (C__11482211_10), and rs13147758 (C___2965080_10). These SNPs are located in intronic (noncoding) regions. Appendix A summarizes the principal characteristics of the assessed SNPs.

Genotyping was evaluated by applying real-time PCR (qPCR) in a StepOne Real-Time PCR System (Applied Biosystems/Thermo Fisher Scientific Inc., Singapore), and genotype assignment was performed by sequence detection software (SDS) version 2.3 (Applied Biosystems, CA, USA).

### 2.7. Serum and Sputum HHIP Protein Level Measurement

The determination of protein levels in serum (*n* = 80) and sputum supernatant samples (*n* = 40) was performed by a commercial ELISA kit (cat. E-EL-H0888. Elabscience, Houston, TX, USA) according to the manufacturer’s specifications. The micro ELISA plate was precoated with a human HHIP-specific antibody (detection range: 0.31 – 20 ng/mL, sensitivity: 0.19 ng/mL), and the assays were performed in duplicate on the same plate.

### 2.8. Statistical Assessment

The demographics, clinical characteristics, pulmonary function data, protein levels, and correlations were described using SPSS v.24.0 (IBM, New York, USA). The median, minimum, and maximum values for each continuous quantitative variable were determined.

Hardy-Weinberg equilibrium (HWE) was calculated before performing genotype analysis using PLINK software v1.9 [17], and De Finetti diagrams were constructed with Finetti software v.3.0.8 [18]. The analysis of the genetic association between groups was evaluated by comparing allele and genotype frequencies through Pearson’s chi-square test and Fisher′s exact test using Epi Info v. 7.1.4.0 [19], Epidat statistical software version 3.1 [20], and the haplotype analysis was performed with Haploview v4.2 [21] and R version 3.6.2 (12 December 2019).

The results were considered to be significant when the *p*-value was <0.05; similarly, the odds ratio (OR) with 95% confidence intervals (CI) was estimated to determine the strength of the association. To adjust for potential confounding variables, a logistic regression analysis was performed using Plink v. 1.09. [17] (1 degree of freedom), including age, body mass index, and biomass-burning smoke exposure index as covariables.

Analysis of the HHIP protein levels in serum and sputum supernatant was performed with R version 3.6.2 (12 December 2019), applying the Kolmogorov–Smirnov test, the Mann–Whitney U test for two group comparisons, the Kruskal–Wallis test for three or more comparisons, and Pearson′s r^2^ value for correlations among protein levels and lung function.

### 2.9. Drugs′ Metabolism in Silico Analysis for COPD and Its Interaction with HHIP

To evaluate the probable effects of pharmacological treatment on HHIP protein levels, a in silico analysis was carried out; First-line drugs were documented from clinical records, identifying the targets, carriers, enzymes, and transporters participating in the drugs′ metabolism for the COPD treatment, using the DRUGBANK v.5.1.7 database [22], released 2 July 2020. The identified proteins were used for the interaction analysis, and HHIP was added in the STRING software v11.0 [23].

## 3. Results

### 3.1. Demographic and Clinical Characteristics

The clinical and demographic characteristics of the patients are outlined in Table 1. The median age in the group of women with COPD was 73 years, while biomass-burning smoke-exposed subjects (BBES) were younger by approximately ten years. The COPD group presented a lower body mass index (26.4) than the control group (27.8); this difference was determined to be significant.

Regarding the biomass-smoke exposure index (BSEI), the COPD group was exposed approximately 100 h more than the control group. As expected, the pulmonary function tests in the COPD group were lower than those in the control group. Regarding the distribution by degrees of GOLD severity, > 80% of the participants were GOLD I and II.

### 3.2. Hardy-Weinberg Equilibrium and Genotype Frequencies and Genetic Susceptibility

The three SNPs that were evaluated comply with the HWE: rs13118928 (*p* = 0.23), rs1828591 (*p* = 0.16), and rs13147758 (*p* = 0.40).

The DNA samples of 743 women with and without COPD who were exposed to biomass-burning smoke were genotyped, and three SNPs (rs13147759, rs1828591, and rs13118928) were evaluated. The relationship between cases and controls is 1 case for 2.99 controls; therefore, the statistical power of our study is 90% using the following parameters: an OR = 2, a ratio of controls to case 3:1, a MAF = 15%, and a confidence interval of 95%.

Table 2 shows the results of the genotype frequencies; in the three polymorphisms, the AA genotype was determined to be the most common genotype, and the GG genotype was the least frequently observed genotype. Interestingly, the heterozygous genotypes in both groups occurred in more than 40% of our study population.

The GG homozygous genotype of rs13118928 shows a decreased risk of developing the disease (*p* = 0.038), and the statistical test used was the χ^2^ test. The analyses with the rs13147758 and rs1828591 polymorphisms did not demonstrate significant differences.

Furthermore, all significant results were adjusted for possible confounding variables (age, BMI, and BSEI) through a logistic regression model, where the rs13118928 GG genotype maintained its association with decreased risk (adjusted *p* = 0.021, OR = 0.51, 95% CI 0.27–0.97).

### 3.3. Genetic Models

The three polymorphisms were analyzed by the codominant and recessive genetic association models; as observed in Table 3, rs13118928, which showed a decreased risk of disease susceptibility in the genotype frequencies, maintained this result with the GG genotype in the recessive model (*p* = 0.038, adjusted *p* = 0.0023 OR = 0.51, 95% CI = 0.27–0.97). The OR and the confidence interval were maintained in both analyses, and the *p*-value was strengthened after the logistic regression analysis, demonstrating that minor allele homozygous carriers have a decreased COPD risk.

### 3.4. Haplotypes

Appendix A presents the haplotypes identified in the rs13147758 and rs1828591 polymorphisms. Four haplotypes were formed, with two showing significant trends: AA (*p* = 0.07, OR = 1.25, 95% CI = 0.98–1.60) and AG (*p* = 0.06, OR = 0.53, 95% CI = 0.27–1.02). Figure 1 shows the haplotypes formed by rs1828591 and rs13118928 (r^2^ = 0.54); the haplotype formed by both minor alleles (G) presented a decreased risk of disease (*p* = 0.04, OR = 0.65, CI95% = 0.43–0.98). In the analysis examining the three SNPs, no haplotypes were determined to be associated.

### 3.5. Severity in COPD and Genetic Analysis

All COPD patients were clustered according to the degree of disease severity following the GOLD guidelines. Patients in GOLD grades I and II were grouped as G1 (*n* = 149), and those in the disease stages GOLD III and IV were grouped as G2 (*n* = 37). No significant differences were found in any of the 3 SNPs after comparing the genotype frequencies between these two groups. These results are shown in Appendix A.

### 3.6. HHIP Serum Levels

The serum HHIP protein levels were determined through ELISA assays in 80 randomly selected women exposed to burning biomass smoke (COPD-BS = 40 and BBES = 40) (Appendix A); these subgroups were derived from the main groups. COPD patients had a median age of 71.5 years, while in women without the disease, the mean age was 64 years (*p* = 0.003); for this reason, age, and BMI, were employed as a covariables to adjust.

#### 3.6.1. HHIP Serum Level Comparison

This analysis was performed by comparing HHIP serum protein levels between both groups, regardless of polymorphism; however, this comparison was not significant (*p* = 0.62) (Figure 2).

The comparison between cases and controls depending on the polymorphism showed significant differences with the SNPs rs13147758 (AA; *p* = 0.021) and rs1828591 (AA; *p* = 0.023); however, after the logistic regression analysis, these associations were not significant. A similar result was observed in the intra-case analysis, as rs13147758 showed a difference between cases and controls with the AA genotype (*p* = 0.021), and after adjustment for covariates, it changed to *p* = 0.28. The effect same occurred with rs1828591 (*p* = 0.046, after adjustment, *p* = 0.053). In this analysis, rs13118928 did not present any significant difference before adjustment for covariates.

#### 3.6.2. Analysis of Serum Protein Levels by Applying Genetic Models

The analysis of genetic association models was performed using the dominant and recessive models, comparing protein levels in both groups.

Among individuals exposed to biomass-burning smoke without the disease (BBES), the dominant model (AA vs. AG + GG) of rs13118928 showed that individuals who carry one or two copies of the minor allele (G) have higher protein levels compared with individuals who are homozygous for the common allele (A) (*p* = 0.005, and after adjustment, *p* = 0.04). However, in the COPD-BS group, significant differences were not observed (Figure 3).

#### 3.6.3. Correlations of Serum Protein Levels in COPD-BS and BBES with Lung Function

The rs1828591 showed a positive correlation among heterozygosity (AG) and the FEV_1_/FVC ratio in the COPD-BS group (*p* = 0.01, r2 = 0.56), and after logistic regression analysis, this value remained significant (*p* = 0.02, r^2^ = 0.52) (Figure 4A).

Two positive correlations were observed in rs13118928: the first in the BBES group with the AG genotype and FEV_1_ (*p* = 0.04, r^2^ = 0.50; after the analysis by covariates, *p* = 0.03, r^2^ = 0.53). In addition, in the COPD-BS group, the heterozygous (AG) genotype showed a positive correlation with FEV_1_/FVC (*p* = 0.006, r^2^ = 0.56 and adjusted values *p* = 0.012 and r^2^ = 0.54) (Figure 4B,C).

### 3.7. HHIP Levels in Supernatant Sputum of Smoke Biomass Burning

Analysis of HHIP protein levels in sputum supernatant (SS) from subjects exposed to smoke by biomass burning was performed in a subgroup of 40 randomly selected participants (20 COPD-BS and 20 BBES).

The age of the COPD-BS group (median = 72) was higher than that of the BBES group (median = 61.5) ~10 years, which was determined to be significant. For BBEI, the COPD group was more heavily exposed with a median = 470 h/year, and the BBES group had a median of 360 h/year; however, this difference was not significant. On the other hand, pulmonary function test results are lower in the COPD-BS group than in the BBES group, and this difference is expected in our study, since lung function tests are the diagnostic criteria (Appendix A).

#### Protein Level Comparison in Sputum Supernatant

In general, when comparing the protein levels between both groups, regardless of the polymorphism, the BBES group had higher protein levels than did the COPD-BS group (*p* = 0.09); however, this difference was not significant (Figure 5).

In the analysis, according to genotypes among case-controls, no differences were found in the general association, genetic models, or correlations.

### 3.8. HHIP Levels and Metabolism of Drugs for COPD

Nine drugs were identified as the first-line treatment (tiotropium bromide, umeclidinium, vilanterol, fluticasone furoate, salbutamol, ipratropium bromide, budesonide, fluticasone, and/or salmeterol). Based on the above, the in silico analysis showed a strong interaction between the different drugs and the enzymes, carriers, targets, and transporters that participate in their metabolism. However, as shown in supplementary Appendix A, no interaction among the drugs used in treating COPD, and the HHIP protein was identified.

## 4. Discussions

COPD is caused by exposure to noxious particles or gases, such as cigarette smoke or smoke caused by the combustion of solid fuels, known as biomass, which may include coal or crop residue, grass, dry branches, animal dung, charcoal, and wood [24,25].

In the current study, three polymorphisms (rs13118928, rs1828591, and rs13147758) in the *HHIP* gene were analyzed in Mexican women exposed to biomass-burning smoke; this kind of environmental risk factor is relatively common in low- and middle-income countries, such as Mexico, in addition to women being more likely than men to be affected by this exposure [26,27]. Additionally, some authors have suggested that biomass-burning smoke has a different pathophysiological mechanism from that caused by cigarette smoke [28].

The clinical and demographic characteristics of our study group are similar to those of previous reports; Moran-Mendoza et al., in 2008, described a group of women affected by exposure to smoke from biomass burning, aged approximately 67 years, mostly residing in rural or suburban areas, who had 5 h/day of direct exposure to smoke for up to 45 years [29]. Additionally, the BSEI may reach 275 h/year [25]. In our current study, the COPD group exceeded 300 h per year for harmful particles.

Previously, two GWAS seeking to identify genetic susceptibility to COPD in the Caucasian population found two SNPs in the *HHIP* gene (rs1828591 and rs13118928) with consistent replications in three cohorts; however, the combined *p*-values did not reach levels of significance at the GWAS level (1.74 × 10^−7^ and 1.67 × 10^−7^) [7]. In addition, in the Boston Early-Onset COPD (BEOCOPD) family cohort, rs1828591 and rs13118928 were associated with FEV_1_ (*p* = 0.0025 and *p* = 0.0014, respectively), but neither SNP was significantly associated with COPD. Additionally, these same polymorphisms were associated with FEV_1_ in the British Birth Cohort (*p* = 0.039 and *p* = 0.038, respectively) [7]. In another GWAS, rs13147758 had a genome-wide *p*-value associated with FEV_1_/FVC and FEV_1_, as well as airflow obstruction, in smokers [8].

Some studies in Asian populations have evaluated these particular polymorphisms in the HHIP gene but focused on their relationship with tobacco smoking; in the KOLD cohort, 15 SNPs were analyzed, and none was associated with COPD; however, four were significantly associated with FEV_1_ value [30]. Similarly, in a Chinese Han population, none of the SNPs in HHIP had an association with COPD, but the rs12509311, rs13118928, and 1,828,591 were associated with the FEV_1_/FVC ratio in COPD smokers [31]. In contrast, three SNPs (rs13147758, rs1828591, and rs13118928) were associated with a decreased COPD risk (OR = 0.57, 0.54 and 0.56, respectively) with the GG genotype in a comparative study between Chinese Han and Mongolian populations [32]. We selected these SNPs because a critical role in the disease has been observed in other populations; however, in biomass-burning smoke-exposed individuals, their participation has not been evaluated to date. In our current study in biomass-burning smoke-exposed women, we found that the rs13118928 GG genotype was associated with a decreased risk (OR = 0.51) of the disease; interestingly, these populations have a higher Amerindian component [33]. Regarding the analysis by genetic association models (recessive model), it was shown that carrying two copies of the minor allele (GG) also provides a decreased risk of disease. In a recent study in Mexican-mestizo smokers, we found an association with COPD susceptibility with rs13147758 [34] but not rs13118928. [33,34].

Interestingly, the GG haplotype (rs13118928–rs1828591) shows that both SNPs′ minor alleles confer a decreased risk of COPD. Previously, we reported that rs13147758 (not associated in the current exposure comparison) and the haplotype formed with rs1828591 are associated with smokers’ COPD susceptibility. In the current study, we did not observe any genetic association with susceptibility at the allele or haplotype level. This lack of association is probably due to the minimal differences in the genotype frequencies among cases and controls (approximately 3% between groups), which is in contrast to the tobacco-smoking study, where the difference in genotype frequencies reached 10%. [33,34].

The *HHIP* gene encodes a protein with the same symbol; this protein is a regulatory component of the Hedgehog pathway for embryonal development in vertebrates. HHIP is a transmembrane protein that attenuates the signals of this pathway and, from an evolutionary perspective, it is highly conserved. HHIP plays an essential role in embryogenesis processes, such as lung development and the development of other organs [11]. In a haploinsufficient murine model exposed to cigarette smoke, the importance of the HHIP protein in lung development was demonstrated; homozygous (Hhip-/-) mice died in the short term after birth due to defects in lung morphogenesis. In contrast, heterozygous mice (Hhip+/−) were viable with normal lung development but with an approximately 33% decrease in protein expression and an increase in functional and histological emphysema [35].

The diagnosis of COPD is performed through the spirometry test, and no other procedures are indicated (such as bronchoalveolar lavage or lung tissue biopsy) [1]. We believe that the biological sample closest to the pulmonary microenvironment is sputum; therefore, in this study, we decided to determine protein levels in sputum supernatant and serum. In addition, these measurements have not been previously reported for this pathology.

Interestingly, most of the results associated with the disease were obtained in serum protein levels; however, comparing cases and controls, regardless of polymorphisms, no significant differences were found.

In the case-control comparison, depending on the genotype/polymorphism, with rs13118928 being associated at the genetic level, no differences at the serum protein level were observed. Interestingly, among non-COPD exposed subjects, according to the dominant model (AA vs. AG + GG), the HHIP serum levels are increased among subjects carrying one or two copies for the minor allele (G). Notably, the GG genotype was associated with decreased risk. In this work, we observe that all subjects with one or two copies of the minor allele have higher levels of the protein, which acquires relevance at the biological level, since previous investigations in murine models observed that the protein may play a protective role against harmful stimuli, such as exposure to cigarette smoke [36]. However, this protective role has not been previously described in people exposed to smoke from biomass burning.

In the lung function correlations with serum protein, the HHIP levels correlated positively with the FEV_1_ and FEV_1_/FVC parameters. Additionally, the rs13118928 AG genotype showed that both in controls with FEV_1_ and in cases with FEV_1_/FVC, lung function was better at higher serum protein levels, suggesting that the protein is stimulated by the oxidative stress caused by noxious particles or gases, such as those contained in the smoke from biomass combustion, similar to that described in murine models of tobacco smoke exposure [36]. Our findings with the AG genotype but not with the GG genotype could be due to the reduced number of subjects carrying the GG genotype.

On the other hand, supernatant sputum protein levels did not exhibit significant differences, either in the first case-control comparison or in subsequent analyses, where comparisons depend on genotypes/SNPs. Notably, the HHIP protein levels in the sputum supernatant were consistently lower than those in serum samples.

According to the databases and software consulted [22,23], no direct or indirect interactions of the different pharmacological drugs with the HHIP protein were identified; this allows us to presume so far HHIP levels are not they are altered by the pharmacological treatment used.

Our study has a number of limitations, among which we consider the small number of sputum supernatant samples obtained for protein level determination; although the statistical power for serum and supernatant protein levels is 68% and 60%, respectively, in a study such as the one that we conducted and in a poorly explored population, this approach presents good statistical power between protein levels and lung function.

Finally, our study analyzed three polymorphisms that had been previously associated with COPD in other populations. However, to the best of our knowledge, no previous association studies have investigated populations exposed to smoke due to biomass burning. Additionally, our study is the first to describe the protein levels in serum and sputum in subjects exposed to smoke by biomass burning with and without COPD. Our results may pave the way for further studies investigating COPD pathophysiology, where the HHIP participates at the gene-variation and protein levels.

## 5. Conclusions

The rs13118928 GG genotype and the rs13118928–rs1828591 (GG) haplotype are associated with decreased COPD risk in Mexican women exposed to smoke from biomass burning.

In addition, the HHIP serum protein levels in subjects harboring the rs13118928 AG genotype exposed to smoke by biomass burning, both with and without COPD, are associated with better lung function.

## Figures and Tables

**Figure 1 diagnostics-10-00734-f001:**
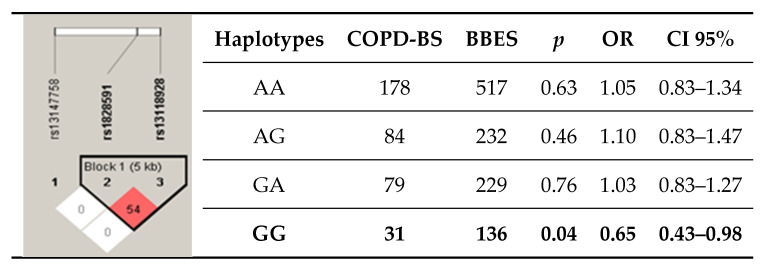
Haplotypes of rs1828591 and rs13118928 in the *HHIP* gene. COPD-BS: COPD related to biomass-burning exposure; BBES: Biomass-burning smoke-exposed subjects: *p* < 0.05 demonstrates significance; OR: odds ratio; CI 95%: 95% confidence interval; showing r^2^ values among SNPs.

**Figure 2 diagnostics-10-00734-f002:**
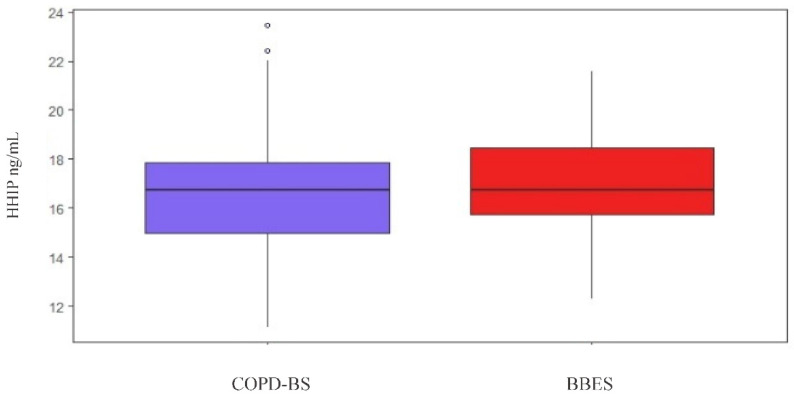
Comparison of protein HHIP in serum levels among COPD-BS vs. BBES.

**Figure 3 diagnostics-10-00734-f003:**
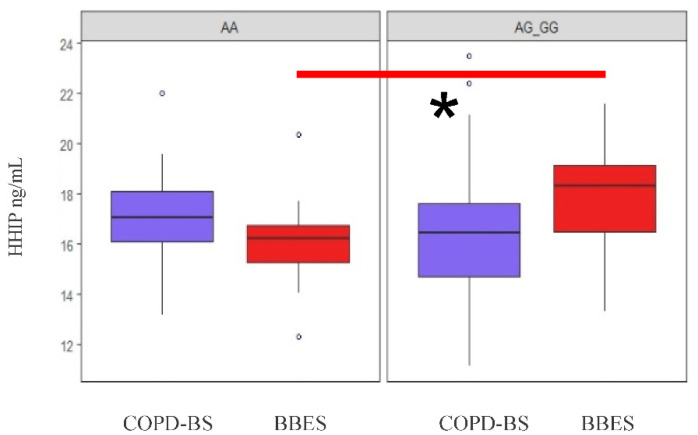
Levels of HHIP protein in serum by the dominant model of rs13118928. * The statistical difference between BBES with AA vs. AG+GG (*p* < 0.05).

**Figure 4 diagnostics-10-00734-f004:**
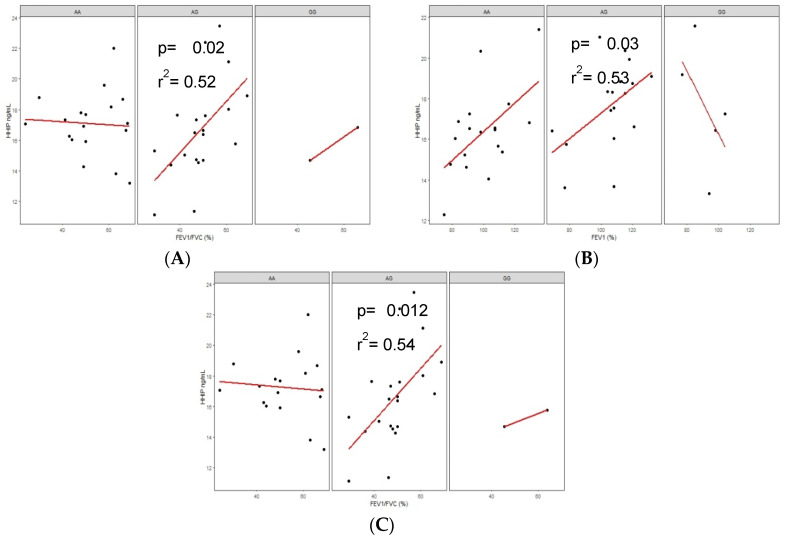
Correlations of protein levels and lung function. (**A**) correlation of FEV_1_/FVC and protein levels in heterozygous (AG) BBES group of the rs1828591; (**B**) correlation of FEV_1_ and protein levels in heterozygous (AG) BBES group; and (**C**) correlation of FEV_1_/FVC and protein levels in heterozygous (AG) COPD-BS group, both in the rs13118928.

**Figure 5 diagnostics-10-00734-f005:**
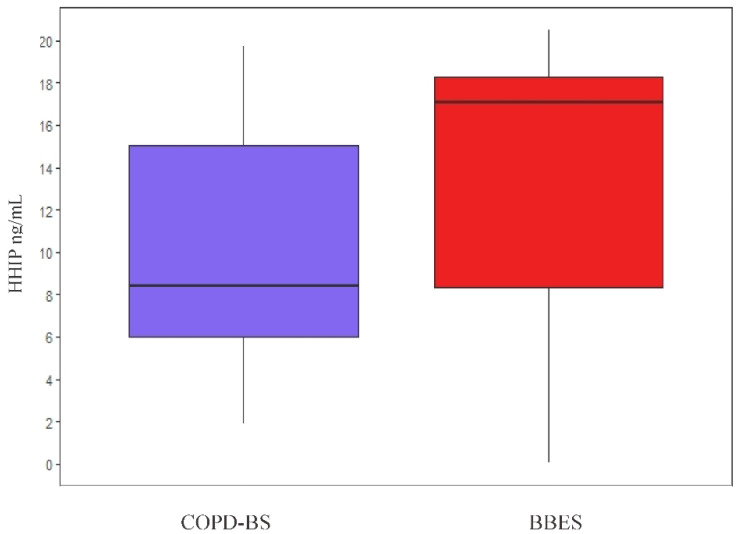
Comparison of HHIP protein in supernatant sputum levels among COPD-BS 4. 3.

**Table 1 diagnostics-10-00734-t001:** Demographic, clinical, exposure and lung function of participants exposed to biomass-burning smoke.

Variables	COPD-BS (*n* = 186)	BBES (*n* = 557)	*p*
Age (Years)	73 (47–93)	62 (45–98)	<0.001
BMI	26.47 (17.81–45.04)	27.83 (15.67–56.25)	0.020
Years of exposure to smoke biomass-burning	50 (10–80)	43 (10–87)	0.005
Hours/day exposure	8 (2–24)	7 (2–24)	0.003
BBEI	320 (108–1116)	240 (105–1050)	<0.001
Lung function post-bronchodilator			
FEV_1_ (%)	64 (18–119)	98 (55–187)	<0.001
FVC (%)	83 (35–146)	94 (53–193)	<0.001
FEV_1_/FVC (%)	58 (21.53–69.7)	83 (70–138)	<0.001
GOLD			
GOLD I (%)	46 (24.7)	NA	-
GOLD II (%)	103 (55.4)	NA	-
GOLD III (%)	27 (14.5)	NA	-
GOLD IV (%)	10 (5.4)	NA	-

COPD-BS: chronic obstructive pulmonary disease (COPD) related to biomass-burning exposure; BBES: biomass-burning smoke-exposed subjects. *p* < 0.05 statistical significance; BMI: body mass index; BBEI: biomass-burning smoke exposure index; FEV_1_: forced expiratory volume in the first second; FVC: forced vital capacity; NA: not applicable. The median and minimum and maximum values are shown.

**Table 2 diagnostics-10-00734-t002:** Genotype frequencies among study groups.

Genotype	COPD-BS		BBES		*p*	OR	CI 95%	* *p*
	*n* = 186	%	*n* = 557	%				
rs13147758								
AA	87	46.78	243	43.63	0.49	0.13	0.89–1.28	
AG	80	43.01	243	43.63	0.93	0.97	0.69–1.36	
GG	19	10.21	71	12.74	0.43	0.80	0.49–1.29	0.42
rs1828591								
AA	90	48.38	257	46.16	0.61	1.09	0.78–1.52	
AG	77	41.41	232	41.64	1	0.98	0.70–1.38	
GG	19	10.21	68	12.20	0.51	0.81	0.47–1.40	0.13
rs13118928								
AA	88	47.32	258	46.32	0.86	1.04	0.74–1.45	
AG	86	46.23	233	41.83	0.30	1.19	0.85–1.66	
**GG**	**12**	**6.45**	66	**11.85**	**0.038**	**0.51**	**0.27–0.97**	**0.021**

COPD-BS: COPD related to biomass-burning exposure; BBES: exposed to biomass-burning smoke. *p* < 0.05 statistical significance; * *p*-value adjusted by age, BMI, and BBEI. Associations are shown in the full genotype model.

**Table 3 diagnostics-10-00734-t003:** Analysis by genetic models of rs13118928 in exposed to biomass-burning smoke with and without COPD.

Model/Genotype	COPD-BS		BBES		*p*	OR	CI 95%	* *p*
	*n* = 186	%	*n* = 557	%				
Codominant								
AA	88	47.32	258	46.32	0.25	1 (ref.)		0.36
AG	86	46.23	233	41.83		1.08	0.76–1.52	
GG	12	6.45	66	11.85		0.53	0.27–1.03	
Recessive								
**GG** **AA + AG**	12	6.45	66	11.85	**0.038**	**0.51**	**0.27–0.97**	**0.0023**
	174	93.55	491	88.15		1.94	1.02–3.69	

COPD-BS: COPD related to biomass-burning exposure; BBES: biomass-burning smoke-exposed subjects. *p* < 0.05 statistical significance; * *p*-value adjusted by age, BMI, and BBEI.

## Data Availability

The datasets generated and analyzed for this study can be found in ClinVar SCV001423136, SCV001423137, and SCV001423138.

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
