# Peer review of "Participation of HHIP Gene Variants in COPD Susceptibility, Lung Function, and Serum and Sputum Protein Levels in Women Exposed to Biomass-Burning Smoke"

_diagnostics, 2020, doi:10.3390/diagnostics10100734_

Round 1

Reviewer 1 Report

As one of the leading cause of death in the world, chronic obstructive pulmonary disease (COPD) has been proven to be strongly associated with cigarette smoke and genetic factors. Hedgehog interacting protein is crucial to distinct aspects of vertebrate development and organ maintenance. The single nucleotide polymorphisms (SNPs) of HHIP gene have been implicated in COPD by multiple genome-wide association studies (GWAS). Though a variety of SNPs in HHIP genes have been studied in regarding to the susceptibility to COPD, the specific SNP in specific ethnic group facing specific environmental risks remain elusive. In developing countries like Mexico, people especially women facing a variety of organic materials (biomass) in poorly ventilated indoor areas on daily basis and smoke from this combustion is also considered as an Ambiental risk factor to COPD. In this manuscript, the authors conducted a case control study of contribution SNPs in the HHIP in Mexican women exposed to biomass-burning smoke with and without COPD. The authors identified several SNPs correlate with decreased COPD risk and better lung function. Moreover, by measuring the HHIP serum, and sputum supernatant levels in Mexican women, the authors also draw the conclusion that HHIP SNPs also correlate with serum protein levels and could be used as an indicator in COPD.

The manuscript was overall well written and the data was convincible and well presented. I believe it is suitable to be published on Diagnosis by addressing several concerns as following:

1. Since this study aimed to investigate the role of SNPs in HHIP on COPD susceptibility in women of single ethnic groups, the authors should be more clear about the Mexico women included as reference 32.

2. According to Table 2 and Figure 1, the GG homozygous genotype of rs13118928 shows a decreased risk of developing the disease, while in the haplotypes with rs13147758, the effect seems to be similar. I think this is worth discussing.

3.With three main phenotypes of the COPD: frequent exacerbator (FE), asthma/COPD overlap (ACO), and emphysema with hyperinflation, it’s better for the authors to re-analyze the association of SNPs with serum and sputum protein levels in different phenotypes rather than general lung function to give a more thorough evaluation.

Author Response

Reviewer 1

Comments and Suggestions for Authors

As one of the leading cause of death in the world, chronic obstructive pulmonary disease (COPD) has been proven to be strongly associated with cigarette smoke and genetic factors. Hedgehog interacting protein is crucial to distinct aspects of vertebrate development and organ maintenance. The single nucleotide polymorphisms (SNPs) of HHIP gene have been implicated in COPD by multiple genome-wide association studies (GWAS). Though a variety of SNPs in HHIP genes have been studied in regarding to the susceptibility to COPD, the specific SNP in specific ethnic group facing specific environmental risks remain elusive. In developing countries like Mexico, people especially women facing a variety of organic materials (biomass) in poorly ventilated indoor areas on daily basis and smoke from this combustion is also considered as an Ambiental risk factor to COPD. In this manuscript, the authors conducted a case control study of contribution SNPs in the HHIP in Mexican women exposed to biomass-burning smoke with and without COPD. The authors identified several SNPs correlate with decreased COPD risk and better lung function. Moreover, by measuring the HHIP serum, and sputum supernatant levels in Mexican women, the authors also draw the conclusion that HHIP SNPs also correlate with serum protein levels and could be used as an indicator in COPD.

The manuscript was overall well written and the data was convincible and well presented. I believe it is suitable to be published on Diagnosis by addressing several concerns as following:

  1. Since this study aimed to investigate the role of SNPs in HHIP on COPD susceptibility in women of single ethnic groups, the authors should be more clear about the Mexico women included as reference 32.

Thank you for your kind observation; Now, lines 83 to 89 describe in detail the criteria used for the selection of participants from the GOLD recommendations for diagnosis to the time of exposure to biomass-burning smoke. Also, these participants are different from those in reference 32 since they are exclusively exposed to biomass-burning smoke.

  1. According to Table 2 and Figure 1, the GG homozygous genotype of rs13118928 shows a decreased risk of developing the disease, while in the haplotypes with rs13147758, the effect seems to be similar. I think this is worth discussing.

We welcome your comments; now, in the discussion section, we addressed this point at lines 345 - 351, where we compare our results of current haplotypes with those previously reported in smokers.

3.With three main phenotypes of the COPD: frequent exacerbator (FE), asthma/COPD overlap (ACO), and emphysema with hyperinflation, it's better for the authors to re-analyze the association of SNPs with serum and sputum protein levels in different phenotypes rather than general lung function to give a more thorough evaluation.

We appreciate your exciting comment; however, in our study, all the participants who had asthma (or others respiratory diseases different to COPD) were excluded from the study; this was thought to avoid bias in terms of lung function altered by other respiratory diseases since it is the first time that these variants have been evaluated in women with COPD secondary to exposure to smoke from biomass burning. Also, as mentioned in the materials and methods section, only women with no history of previous exacerbations were included. Besides, as is stated in the same section, all participants were recruited in rural areas of Mexico through campaigns in rural localities, unfortunately, according to the demographical characteristics (mainly rural) of these women, an HRTC nor DLCO is not possible to do.

Reviewer 2 Report

I have read the article by Ortega-Martínez et al. with great interest. The authors investigated HHIP in Mexican patients with COPD. The article has definite merit; however, it is very difficult to read due to grammatical errors and to be honest I did not get the meaning of some sentences. I would strongly encourage the authors to check it by a native speaker, as I feel that some of the important messages may have been misunderstood.

Comments:

  • Please, explain the abbreviation of HHIP in the title and abstract.
  • Methods: GOLD is not a guideline! Please change it to recommendations.
  • Methods: To my experience many ELISA-s are affected by DTT, compromising their performance to quantify biomarker levels. I would be grateful if you could provide figures that was not the case for HHIP.
  • Please identify the main outcome and provide a priori power calculations how was this achieved. Then provide sensitivity power calculations what was the effect size you could detect with the selected sample size for the secondary or observational outcomes.  
  • There were a few differences in lung function variables between the two groups. I wonder if the differences in serum and sputum HHIP levels were adjusted for these covariates.
  • Please provide the list of anti-COPD medications. Could they be responsible for observed differences in the protein levels?
  • Why were ELISA analyses performed only in a subgroup of patients? How were they selected? How are these subgroup analyses powered?
  • At limitation section the authors should explain why it was problematic that the number of patients with protein analyses was low (false conclusions, etc.).

Author Response

Comments and Suggestions for Authors

I have read the article by Ortega-Martínez et al. with great interest. The authors investigated HHIP in Mexican patients with COPD. The article has definite merit; however, it is very difficult to read due to grammatical errors and to be honest I did not get the meaning of some sentences. I would strongly encourage the authors to check it by a native speaker, as I feel that some of the important messages may have been misunderstood.

Thank you very much for your important comment; attending it, we have employed a specialized service to style English language correction (AJE). The current version was amended through all manuscript, including the title. We are adding the certificate for that service.

Comments:

Please, explain the abbreviation of HHIP in the title and abstract.

Thank you for your comment; we have corrected the abstract (see line 29); however, we think that the title is not necessary since it is an official gene symbol. Besides, this change makes most extensive our already long manuscript title.

Methods: GOLD is not a guideline! Please change it to recommendations.

Attending your comments, we have decided to change the term "guidelines" for "recommendations"; this modification can be seen in the manuscript, line 78.

Methods: To my experience many ELISA-s are affected by DTT, compromising their performance to quantify biomarker levels. I would be grateful if you could provide figures that was not the case for HHIP.

We appreciate your comment, and at the same time, we apologize for a mistake in the redaction, because after a reflection about the workflow, part of the sputum sample process was to obtain the cells for which DTT was used; however, this reagent was not used for obtaining the sputum supernatant as explained now in the Methods section (lines 131 - 137). Finally, the cells were not employed in the current study, and the obtention procedure was deleted for the current version.

Please identify the main outcome and provide a priori power calculations how was this achieved. Then provide sensitivity power calculations what was the effect size you could detect with the selected sample size for the secondary or observational outcomes.

We appreciate your comment and suggestion; due to an involuntary omission on our part, the statistical power calculation for the protein analysis was not placed; however, for the group of serum protein levels, it is 68%, while for the sputum supernatant, it is 60%, this result is addressed as a potential limitation at the end of the discussion section, please note that previously we have included the statistical power for genetic association analysis, which reaches around 90%; in both cases are now stated in the manuscript current version. (see lines 392 to 395)

There were a few differences in lung function variables between the two groups. I wonder if the differences in serum and sputum HHIP levels were adjusted for these covariates.

Since lung function parameters are the clinical criteria to establish the diagnosis, these differences are expected in this kind of case-control studies, so they are not taken as possible confounding variables. The results are not adjusted for lung function.

Please provide the list of anti-COPD medications. Could they be responsible for observed differences in the protein levels?

According to the approved international guidelines (ATS/ERS), the pharmacological COPD treatment in our institution is dependent on the GOLD stages being LABA, LAMA, and ICS (either alone or in combination) the first line of treatment. However, there is no scientific information regarding the possible influence over the HHIP protein; finally, we did not evaluate the interaction between the protein and the different treatments since this is not our study's aim.

Why were ELISA analyses performed only in a subgroup of patients? How were they selected? How are these subgroup analyses powered?

The protein was measured in a subgroup since all participants met the same inclusion criteria, so these were randomly selected from both groups; in these analyses, it is sought that a representative group of the total sample reflects the characteristics of our total sample.

At limitation section the authors should explain why it was problematic that the number of patients with protein analyses was low (false conclusions, etc.).

Thank you, now at the end of the Discussion section, this was stated.

Round 2

Reviewer 2 Report

I am happy to see that there were some improvements in the manuscript. However, the authors should address my previous question on the medications and their potential effect.

This is vital as we do not know if the differences in serum and sputum protein levels are caused by COPD or COPD-medications.

Round 3

Reviewer 2 Report

I am happy with the changes and suggest acceptance.

This manuscript is a resubmission of an earlier submission. The following is a list of the peer review reports and author responses from that submission.